# Learning to reason iteratively and parallelly for visual reasoning

## Abstract

Iterative step-by-step computation is beneficial for multi-step reasoning scenarios wherein individual operations need to be computed, stored and recalled dynamically (e.g. when computing the query *"determine color of pen to left of the child in red t-shirt sitting at the white table"*). Conversely, parallel computation is beneficial for executing operations that are mutually-independent and can be executed simultaneously and not necessarily sequentially (e.g. when counting individual colors for the query: *"determine the maximally occuring color amongst all t-shirts"*). Accordingly, in this work, we introduce a novel fully neural *iterative and parallel reasoning mechanism* (IPRM) that combines the benefits of iterative computation with the ability to perform distinct operations simultaneously. Our experiments on various visual question answering and reasoning benchmarks indicate that IPRM exhibits stronger reasoning capabilities and generalization than existing recurrent as well as transformer-based reasoning and vision-language interaction mechanisms while requiring lesser parameters and computation steps. Notably, IPRM achieves state-of-the-art zero-shot performance on the challenging CLEVR-Humans dataset and outperforms prior task-specific methods for the NLVR and CLEVR-CoGen benchmarks. Further, IPRM's computation can be visualized across reasoning steps aiding interpretability and diagnosis of its reasoning and outputs.

## 1 Introduction

Visual reasoning and question answering (VQA) at its core requires a model to identify relevant visual operations and accordingly execute and compose their results to make an inference. Iterative computation, wherein individual operations are identified and composed in a step-by-step manner has been shown to be an effective visual reasoning mechanism (Hudson & Manning, 2018; Chen et al., 2018; Vaishnav & Serre, 2022). Notably, the MAC architecture (Hudson & Manning, 2018) captures this computation through a recurrently operated memory-attention-and-control cell, and demonstrates impressive performance and interpretability on the CLEVR (Johnson et al., 2017a) VQA dataset. However, iterative computation while effective, is not necessarily always optimal. For example, a model that only performs iterative reasoning may end up learning more complex or overly task-tuned reasoning procedures than is required, which might lead to it having poor generalization or adaptation to unseen scenarios and newer tasks.

In many scenarios, the entailed intermediate visual operations are mutually independent and it can be more optimal, from both an efficiency and efficacy perspective, to compute them simultaneously instead of iteratively. For example, consider the the first scenario shown in fig. 1. When executing the language phrase *"maximum occuring shape"* (i.e. *"what shape appears the most"*), a purely iterative method would: (i) compute the count of each shape (each of which itself could take multiple iterations), (ii) then update and maintain the counts in memory (without forgetting count of all previous shapes), and (iii) finally, recall each shape's count to compute the *"maximum"* [1]. Besides taking more reasoning steps than optimal, such computation also increases the demand for information retention and recall in memory, which in this scenario could scale by the number of shapes to be counted.

---

[1] Assuming that the *"maximum"* operation is applied only after the counts of all shapes have been computed.

Figure 1: Reasoning scenarios (CLEVR-Humans (Johnson et al., 2017b), GQA (Hudson & Manning, 2019b) and NLVR (Suhr et al., 2017)) wherein combination of iterative (step-by-step) computation (blue phrases in image) and parallel computation of **mutually-independent** operations (orange phrases in image) can be more effective than either of them individually.

An alternative mechanism is to parallelly (simultaneously) compute the counts of each shape since these computations are mutually independent. Thereafter, the computed results can be compared to evaluate the *"maximum"* operation. The same mechanism could also be beneficial for the other two scenarios illustrated in fig. 1. *"Are x and y both made of plastic"* could involve simultaneous execution of *"x made of plastic"* and *"y made of plastic"*, followed by *"and"* composition. Meanwhile, for scenario 3, the condition 1 (*"both images have atleast.."*) of *'and'* is independent of condition 2 (*"only one image.."*) and thus could be computed for each image simultaneously (instead of processing each condition and image in separate steps).

Parallel computation can be realized in conventional transformer-based attention mechanisms (Vaswani et al., 2017). Specifically, these mechanisms involve multiple query tokens simultaneously attending to key-value tokens which effectively capture multiple query-key interactions in a single parallel computation. However, transformer-based attention does not explicitly incorporate iterative compositional computation which, as mentioned before, can be useful for composing intermediate operations. Hence, while transformer mechanisms may effectively compute the result of *"maximum occuring shape"* in fig.1, they would potentially struggle to integrate the result with further operations such as *"green object with .."*, *"small object in front of green .."*, and *"color of .."* that need to be computed step-by-step to answer the question. This capability could be effectively achieved by combining iterative computation with parallel computation.

Based on the above insights, we seek to develop a neural reasoning architecture that combines step-by-step iterative computation with the the ability to perform multiple independent operations simultaneously. Considering scenario 1 again, the architecture would ideally first perform *"maximum occurring shape"* by executing multiple distinct "count" queries for each shape and storing these results in memory. In its next reasoning step, it would perform the *"maximum"* operation by recalling its prior results and composing them conditionally. Thereafter, over subsequent reasoning steps, it would perform necessary composition for '*green object.."*, *"small object in front.."* and *"color of.."* to finally determine the answer. Further, for each computation step, it would also highlight which operations it executes and accordingly where it looks visually.

We accordingly design a novel *iterative* and *parallel reasoning* mechanism (IPRM) which models memory as a set of latent operation states, keyed to which are result states. Given visual and language features as inputs, IPRM performs the following iterative computation. First, it forms a set of new latent operations by retrieving information from the input language features, conditioned on its current operation states. Then, it *"executes"* these latent operations by retrieving relevant visual information conditioned on both the operations and current result states. Finally, it integrates these new latent operations (and their results) into memory by dynamically composing the operations with other simultaneous operations as well as prior operation states. This strategy effectively enables us to take advantage of both parallel and iterative computations and helps improve the performance across various visual reasoning tasks using a single reasoning mechanism. These include compositional visual question answering (using GQA (Hudson & Manning, 2019b) and CLEVR), reasoning generalization to new language forms (CLEVR-Humans), language-grounded visual reasoning (NLVR) and compositional reasoning generalization (CLEVR-CoGen (Johnson et al., 2017a)).

**Contributions:** (i) We introduce a novel iterative- and parallel-reasoning mechanism by drawing insights from the benefits and limitations of both purely iterative reasoning and purely parallel computation. (ii) We demonstrate that our mechanism outperforms existing reasoning and vision-language interaction mechanisms on multiple benchmarks while requiring fewer parameters and computation steps. (iii) Our mechanism exhibits strong generalization on unseen reasoning scenarios and better transfer to new reasoning tasks, notably achieving state-of-the-art zero-shot performance on the CLEVR-Humans benchmark.

## 2 ITERATIVE AND PARALLEL REASONING MODULE (IPRM)

Our proposed iterative- and parallel-reasoning mechanism (IPRM) is a fully-differentiable neural architecture. Given visual features $\mathbf{X_V} \in \mathbb{R}^{N_V \times D_V}$ and language or task-description features $\mathbf{X_L} \in \mathbb{R}^{N_L \times D_L}$, IPRM outputs a *"reasoning result"* $\mathbf{y_s} \in \mathbb{R}^{D_m}$ and, optionally, a set of *"reasoning result tokens"* $\mathbf{Y_R} \in \mathbb{R}^{N_m \times D_m}$. As previously mentioned, IPRM operates iteratively for $T$ reasoning steps and internally, maintains an explicit memory $\mathbf{M} : \{\mathbf{M_{op}}, \mathbf{M_{res}}\}$. The memory is modelled as a set of *"operation states"* $\mathbf{M_{op}} \in \mathbb{R}^{N_m \times D_m}$, keyed to which are *"result states"* $\mathbf{M_{res}} \in \mathbb{R}^{N_m \times D_m}$ as shown in fig. 3. Here, $N_m$ denotes the number of parallel operations to be computed while $D_m$ denotes the mechanism's internal feature dimension. On a high level, at each reasoning step (denoted by $t \in \{1, \cdots, T\}$), IPRM performs the following computations:

1. First, conditioned on its existing operation states $\mathbf{M_{op,t}}$, it retrieves relevant information from $\mathbf{X_L}$ to form a new set of latent operations $\mathbf{Z_{op,t}} \in \mathbb{R}^{N_m \times D_m}$. We term this computation as *"Operation Formation"*.

$$\mathbf{Z_{op,t}} = \mathbf{Operation\_Formation}(\mathbf{X_L}; \mathbf{M_{op,t}}) \tag{1}$$

2. Then, conditioned on $\mathbf{Z_{op,t}}$ and its existing results state $\mathbf{M_{res,t}}$, it retrieves relevant information from $\mathbf{X_V}$ which represent a new set of latent results $\mathbf{Z_{res,t}} \in \mathbb{R}^{N_m \times D_m}$ corresponding to $\mathbf{Z_{op,t}}$. We term this computation as *"Operation Execution"*.

$$\mathbf{Z_{res,t}} = \mathbf{Operation\_Execution}(\mathbf{X_V}; [\mathbf{Z_{op,t}}, \mathbf{M_{res,t}}]) \tag{2}$$

3. Finally, each operation $\mathbf{Z_{op\_k,t}}$ (where $k \in \{1..N_m\}$), is composed with other operations in $\mathbf{Z_{op\_k,t}}$ as well as prior operation states $\mathbf{M_{op[t-W:t]}}$ within a lookback-window $W$. The corresponding result $\mathbf{Z_{res\_k,t}}$ is similarly combined with other results $\mathbf{Z_{op,t}}$ and prior result states $\mathbf{M_{res[(t-W):t]}}$. We term this computation as *"Operation Composition"*

$$\mathbf{M_{t+1}} = \mathbf{Operation\_Composition}(\{\mathbf{Z_{op,t}}, \mathbf{Z_{res,t}}\}, \mathbf{M_{[(t-W):t]}}) \tag{3}$$

As shown in eq. 3, this output is the new memory state $\mathbf{M_{t+1}} : \{\mathbf{M_{op,t+1}}, \mathbf{M_{res,t+1}}\}$.

The overall computation flow is illustrated in fig. 3, and we provide specific details and intuitions behind these computations in the following sub-sections.

### 2.1 OPERATION FORMATION

The *"operation formation"* stage conceptually models a reasoner that based on its prior set of operations, decides what language features to retrieve in order to form the next set of relevant operations. This can be effectively implemented through conventional attention mechanisms. Specifically, the cumulative set of prior operations (maintained in $\mathbf{M_{op,t}}$) can be projected to form the 'query' $\mathbf{Q_{L,t}} \in \mathbb{R}^{N_m \times D_m}$ representing "what features to look for". The language features $\mathbf{X_L}$ can be projected to form the 'key' $\mathbf{K_L} \in \mathbb{R}^{N_L \times D_m}$ and 'value' $\mathbf{V_L} \in \mathbb{R}^{N_L \times D_m}$. Finally, the new set of latent operations $\mathbf{Z_{op,t}}$ can be retrieved by computing `attention`$(\mathbf{Q_L}, \mathbf{K_L}, \mathbf{V_L})$. These steps are formally represented below:

$$\mathbf{Q_{L,t}} = \mathbf{W_{L,q2}}(\texttt{lang\_nonlin}(\mathbf{W_{L,q1}}(\mathbf{M_{op,t}}))) \tag{4}$$

$$\mathbf{K_L}, \mathbf{V_L} = \mathbf{W_{L,k}}(\mathbf{X_L}), \mathbf{W_{L,v}}(\mathbf{X_L}) \tag{5}$$

$$\mathbf{Z_{op,t}} = \texttt{attention}(\mathbf{Q_{L,t}}, \mathbf{K_L}, \mathbf{V_L}) \tag{6}$$

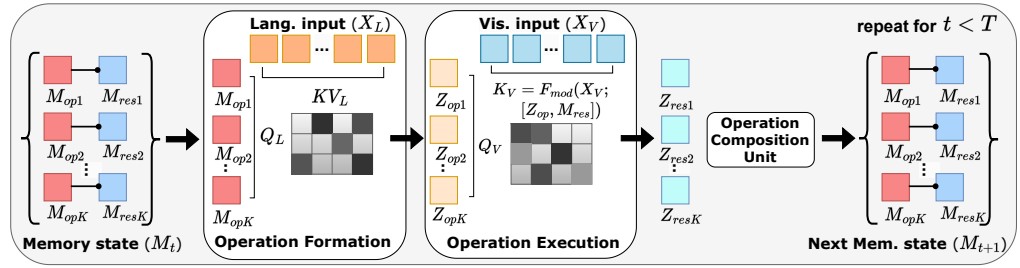

Figure 3: IPRM's computation flow diagram. First, a new set of K-parallel latent operations $\mathbf{Z_{op}}$ are retrieved from language features $\mathbf{X_L}$ conditioned on prior operation states $\mathbf{M_{op}}$. Then, visual features $\mathbf{X_V}$ are queried conditioned on both $\mathbf{Z_{op}}$ and prior result states results $\mathbf{M_{res}}$, to form the new results $\mathbf{Z_{res}}$. Finally, both $\mathbf{Z_{res}}$ and $\mathbf{Z_{op}}$ are passed to the Operation Composition Unit (see sec. 3), the output of which becomes the new memory state $\mathbf{M}$.

Here, $\mathbf{W_{L,q2}} \in \mathbb{R}^{D_m \times D_m}$, $\mathbf{W_{L,q1}} \in \mathbb{R}^{D_m \times D_m}$, $\mathbf{W_{L,k}} \in \mathbb{R}^{D_m \times D_l}$ and $\mathbf{W_{L,v}} \in \mathbb{R}^{D_m \times D_l}$. We set lang_nonlin to GELU for transformer-based language backbones and Tanh for LSTM-based backbones. Note that since $\mathbf{K_L}$ and $\mathbf{V_L}$ are not computation-step dependent, they are computed only once, and only if the input language backbone features are of a different dimension than the internal dimension $D_m$. Regarding attention computation, as shown in eq. 7, it can be implemented as dot-product attention or linear-modulated attention (with appropriate broadcasting and projection weight $\mathbf{W_a} \in \mathbb{R}^{D_k \times 1}$).

$$\texttt{attention}(Q, K, V) = \begin{cases} \text{softmax}\left(\frac{QK^T}{\sqrt{d_k}}\right) V & \text{if dot product attention} \\ \text{softmax}(\mathbf{W_a}(Q \odot K))V & \text{if modulated attention} \end{cases} \tag{7}$$

## 2.2 OPERATION EXECUTION

In the *"operation execution"* stage, the reasoner determines what visual features need to be retrieved depending on both the newly formed operations and existing result states. To model the constituent visual attention mechanism, we draw insights from existing recurrent visual reasoning methods (Hudson & Manning, 2018; Vaishnav & Serre, 2022) that incorporate feature modulation for memory-guided attention. Specifically, in our formulation, we first retrieve a set of feature modulation weights $\mathbf{S_{V,t}} \in \mathbb{R}^{N_m \times D_m/r}$ through a joint projection of the new operations $\mathbf{Z_{op,t}}$ and prior results $\mathbf{M_{res,t}}$ as shown in eq. 8. Here, $r$ is a feature reduction ratio (Hu et al., 2018), $\mathbf{S_{V,t}}$ is then applied dimension wise to a projection of $\mathbf{X_V}$ to retrieve an intermediate attention key $\mathbf{K'_{V,t}} \in \mathbb{R}^{N_m \times N_k \times D_m/r}$. The final attention key $\mathbf{K_{V,t}}$ is then obtained through a joint multi-layer-projection of $\mathbf{K'_{V,t}}$ and the previously projected representation of $\mathbf{X_V}$ as shown in eq. 10. Finally, the attention query and value are formed through separate projections of $\mathbf{Z_{op,t}}$ and $\mathbf{X_V}$ respectively. These are then fed together with $\mathbf{K_{V,t}}$ to the attention function to retrieve the new operation results $\mathbf{Z_{res,t}}$ as shown in eq. 12. Intuitively, the overall process allows for both prior results and the new set of operations to jointly guide visual attention.

$$\mathbf{S_{V,t}} = \mathbf{W_{V,s}}([\mathbf{W_{V,op}}(\mathbf{Z_{op,t}}), \mathbf{W_{V,res}}(\mathbf{M_{res,t}})]) \tag{8}$$

$$\mathbf{K'_{V,t}} = \mathbf{S_{V,t}} \odot \mathbf{W_{V,k1}}(\mathbf{X_V}) \tag{9}$$

$$\mathbf{K_{V,t}} = \mathbf{W_{V,k3}}(\texttt{vis\_nonlin}(\mathbf{W_{V,k2}}([\mathbf{W_{V,k1}}(\mathbf{X_V}), \mathbf{K'_{V,t}}]))) \tag{10}$$

$$\mathbf{Q_{V,t}}, \mathbf{V_{V,t}} = \mathbf{W_{V,q}}(\mathbf{Z_{op,t}}), \mathbf{W_{V,v}}(\mathbf{X_V}) \tag{11}$$

$$\mathbf{Z_{res,t}} = \texttt{attention}(\mathbf{Q_{V,t}}, \mathbf{K_{V,t}}, \mathbf{V_{V,t}}) \tag{12}$$

Here, $\mathbf{W_{V,op}} \in \mathbb{R}^{D_m/r \times D_m}$, $\mathbf{W_{V,res}} \in \mathbb{R}^{D_m/r \times D_m}$, $\mathbf{W_{V,s}} \in \mathbb{R}^{D_m/r \times 2D_m/r}$, $\mathbf{W_{V,k1}} \in \mathbb{R}^{D_m/r \times D_v}$, $\mathbf{W_{V,k2}} \in \mathbb{R}^{D_m/r \times 2D_m/r}$, $\mathbf{W_{V,k3}} \in \mathbb{R}^{D_m/r \times D_m/r}$, $\mathbf{W_{V,q}} \in \mathbb{R}^{D_m/r \times D_m}$ and $\mathbf{W_{V,v}} \in \mathbb{R}^{D_m \times D_v}$. We set vis_nonlin to GELU for transformer visual backbones and ELU for convolutional backbones.

## 2.3 OPERATION COMPOSITION

Finally, in the *"operation composition"* stage, the reasoner first integrates the executed operations $\mathbf{Z_{op,t}}$ and their results $\mathbf{Z_{res,t}}$ into the existing memory state $\mathbf{M_t}$ through a simple recurrent update as shown in eq. 13 and 14. Then, it dynamically composes individual operation states $\mathbf{M'_{op,t+1}}$ with other operation states in $\mathbf{M'_{op,t+1}}$ and also prior operation states in $\mathbf{M_{op,t-W:t}}$ where $W$ is an attention look-back window. This composition is achieved through computing inter-operation attention. Specifically, $\mathbf{M'_{op,t+1}}$ is projected to obtain a set of queries $\mathbf{Q_{op,t}}$, while the token-wise concatenation of $\mathbf{M'_{op,t+1}}$ and $\mathbf{M_{op,t-w:t}}$ are projected to obtain the operation attention keys $\mathbf{K_{op,t}}$ and values $\mathbf{V_{op,t}}$. A second set of values $\mathbf{V_{res,t}}$ are also formed through projection of respective result states as shown in eq. 18. Further, an identity attention mask $\mathbf{I_{N_m}}$ is used to ensure that operations in $\mathbf{Q_{op,t}}$, can only attend to other operations and not themselves. This is done to enable a higher degree of operation composition. As shown in eq. 19, $\mathbf{Q_{op,t}}$, $\mathbf{K_{op,t}}$, $\mathbf{V_{op,t}}$ and $\mathbf{I_{N_m}}$ are passed to the attention operation, which outputs an intermediate representation $\mathbf{M''_{op,t+1}}$ and the softmaxed-attention weights $\mathbf{A_{op,t}}$. $\mathbf{M''_{op,t+1}}$ is subsequently added to a projection of $\mathbf{M'_{op,t+1}}$ to effectively combine attended operation states with the original operation states, and thereby form the next memory operation state $\mathbf{M_{op,t+1}}$. Finally, the next result states are obtained by applying $\mathbf{A_{op,t}}$ on $\mathbf{V_{res,t}}$ and then adding a projection of $\mathbf{M'_{res,t+1}}$ as shown in eq. 21. Note $\mathbf{A_{op,t}}$ is specifically utilized to ensure that results are composed based on attentions between operation states.

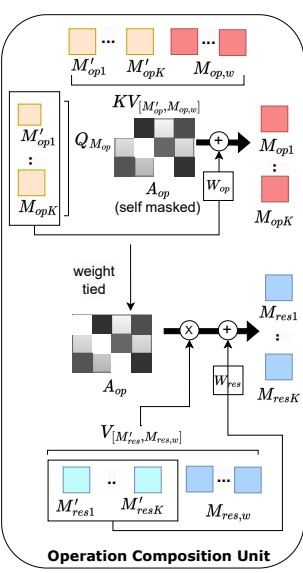

Figure 2: The Operation Composition Unit where operations and their results are composed together to form the new memory state $\mathbf{M_{t+1}}$.

$$\mathbf{M'_{op,t+1}} = \mathbf{W_{op,U}}(\mathbf{Z_{op,t}}) + \mathbf{W_{op,H}}(\mathbf{M_{op,t}}) \tag{13}$$

$$\mathbf{M'_{res,t+1}} = \mathbf{W_{res,U}}(\mathbf{Z_{res,t}}) + \mathbf{W_{res,H}}(\mathbf{M_{res,t}}) \tag{14}$$

$$\mathbf{Q_{op,t}} = \mathbf{W_{op,q}}(\mathbf{M'_{op,t+1}}) \tag{15}$$

$$\mathbf{K_{op,t}} = \mathbf{W_{op,k}}([\mathbf{M'_{op,t+1}}, \mathbf{M_{op,t-W:t}}]) \qquad \text{(token-wise concat)} \tag{16}$$

$$\mathbf{V_{op,t}} = \mathbf{W_{op,v}}([\mathbf{M'_{op,t+1}}, \mathbf{M_{op,t-w:t}}]) \qquad \text{(token-wise concat)} \tag{17}$$

$$\mathbf{V_{res,t}} = \mathbf{W_{res,v}}([\mathbf{M'_{res,t+1}}, \mathbf{M_{res,t-W:t}}]) \qquad \text{(token-wise concat)} \tag{18}$$

$$\mathbf{M''_{op,t+1}}, \mathbf{A_{op,t}} = \texttt{attention}(\mathbf{Q_{op,t}}, \mathbf{K_{op,t}}, \mathbf{V_{op,t}}, \text{mask=}\mathbf{I_{N_m}}) \tag{19}$$

$$\mathbf{M_{op,t+1}} = \mathbf{M''_{op,t+1}} + \mathbf{W_{op,u2}}(\mathbf{M'_{op,t+1}}) \tag{20}$$

$$\mathbf{M_{res,t+1}} = \mathbf{A_{op,t}}(\mathbf{V_{res,t}}) + \mathbf{W_{res,v2}}(\mathbf{M'_{res,t+1}}) \tag{21}$$

Here, all the mentioned weights $\mathbf{W_{..}} \in \mathbb{R}^{D_m \times D_m}$. $\mathbf{I_{N_m}}$ in eq. 19 is an identity matrix which is concatenated with zeros for the window tokens if window length $w > 0$. The operation composition unit is illustrated in fig. 2.

## 2.4 OBTAINING REASONING SUMMARY REPRESENTATION

As mentioned before, our proposed mechanism outputs a set of *"reasoning result tokens"* $\mathbf{Y_R}$ and a *"reasoning result"* $\mathbf{y_s}$. $\mathbf{Y_R}$ is simply equivalent to the last memory result states $\mathbf{M_{res,T+1}}$. To obtain $\mathbf{y_s}$, we perform attention on the last operation states $\mathbf{M_{op,T+1}}$ by utilizing a summary representation $\mathbf{l_s} \in \mathbb{R}^{D_l}$ of $\mathbf{X_L}$ as the attention-query. We set $\mathbf{l_s}$ to be the first language token in case of transformer-based language backbones and as the last hidden state in case of LSTM-based language backbones. As shown in eq. 22, $\mathbf{l_s}$ is projected to obtain a single-token attention query $\mathbf{p_q}$

while $\mathbf{M_{op,T+1}}$ is projected to obtain the attention keys $\mathbf{P_k}$. The attention value is simply the result states $\mathbf{M_{res,T+1}}$, and the output of the attention function is the *"reasoning result"*. Intuitively, this computation corresponds to the reasoner deciding which final operation states in $\mathbf{M_{op,T+1}}$ are most relevant to the summary of the input language or task-description $\mathbf{X_L}$, based on which the corresponding result states $\mathbf{M_{res,T+1}}$ are weighted and retrieved.

$$\mathbf{p_q}, \mathbf{P_k} = \mathbf{W_{pq,q}}(\mathbf{l_s}), \mathbf{W_{pk,k}}(\mathbf{M_{op,T+1}}) \tag{22}$$

$$\mathbf{y_s} = \mathtt{attention}(\mathbf{p_q}, \mathbf{P_k}, \mathbf{M_{res,T+1}}) \tag{23}$$

$$\tag{24}$$

Here, $\mathbf{W_{pq,q}} \in \mathbb{R}^{D_m \times D_l}$ and $\mathbf{W_{pk,k}} \in \mathbb{R}^{D_m \times D_m}$.

## 2.5 REASONING MECHANISM APPLICABILITY AND BENEFITS

Our proposed iterative and parallel reasoning mechanism is weight-tied which means it does not involve computation-step specific parameters. Hence, its computation steps and attention-window can be varied during inference or when transferring models across tasks of different reasoning complexities. Further, as IPRM is end-to-end trainable, it can directly learn to flexibly optimize how many parallel and iterative computations are performed, and the degree to which each contributes to the final reasoning output. For example, on tasks that require a higher degree of step-by-step processing, IPRM can learn to utilize more iterative computations. On the other hand, for tasks that require many simultaneous computations, IPRM can optimize to perform more parallel computations. Additionally, the presence of both computation modes reduces the chances of overly task-specific tuning of the reasoning process, and aids the learning of relatively more general reasoning procedures. This helps improve generalization to unseen reasoning scenarios and transfer to newer reasoning tasks.

Finally, while we study and propose IPRM in the context of visual reasoning, we note that it can be interpreted as a general reasoning process applicable for reasoning tasks beyond visual reasoning. Simply, its inputs $\mathbf{X_L}$ can be generally interpreted as a reasoning task specification (e.g. question, task details, entailment statement) while $\mathbf{X_V}$ can be interpreted as the reasoning stimuli (e.g. images, embodied scenes, language documents, etc). The reasoning process can then operate iteratively and parallelly as described above to obtain the reasoning outputs $\mathbf{y_s}$ and $\mathbf{Y_R}$.

## 3 EXPERIMENTS

We evaluate IPRM on five standard benchmarks: GQA (Hudson & Manning, 2019b) and CLEVR (Johnson et al., 2017a) for compositional visual question-answering, CLEVR-Humans (Johnson et al., 2017b) for assessing reasoning generalizability on free-form questions, NLVR (Suhr et al., 2017) for language grounded visual reasoning and CLEVR-CoGen (Johnson et al., 2017a) for assessing compositional generalization. We adopt three prominent visual reasoning and vision-language interaction mechanisms as baselines. These are: i) Cross-Attention (wherein language operates as the query to key-and-value visual features) as used in BLIP (Li et al., 2022; 2023) and Flamingo (Alayrac et al., 2022), ii) Concat-Attention (wherein language and visual tokens are concatenated and fed through transformer blocks) as used in VILT (Kim et al., 2021) and MDETR (Kamath et al., 2021), and iii) MAC (Hudson & Manning, 2018) – a prominent recurrent memory attention block for visual reasoning. For Cross-Attention and Concat-Attention, we stack multiple layers together for sequential computation, while for MAC we operate it recurrently for a predefined number of steps. Experiment and model implementation details are available in appendix. We also report comparison against benchmark-specific state-of-the-arts in appendix table . 5). Source code for experiments and visualization and pretrained models will be made publicly available via Github.

## 3.1 COMPOSITIONAL VISUAL QUESTION ANSWERING

We first perform analysis on the GQA and CLEVR datasets. These datasets were specifically chosen as they comprise questions that require multi-step compositional reasoning and can serve as useful tests for compositional reasoning capabilities. On both CLEVR and GQA, we perform experiments with MAC, 2-layer Cross-Att, 4-layer Cross-Att, 2-layer Cross-Att blocks and 4-layer Cross-Att

Table 1: Compositional Visual Question Answering (GQA and CLEVR). The indicated parameter count is only for the respective reasoning mechanism and the vision and language backbones are kept the same for fair comparison.

| Mechanism | Param. | GQA-TestDev | | | | | | CLEVR |
|---|---|---|---|---|---|---|---|---|
| | | Ovr. | Query | Verify | Logic | Choose | Compare | |
| MAC | 5.8M | 57.6 | **44.9** | 78.2 | 68.7 | 74.0 | 59.4 | **98.9** |
| Cross-Att (4L) | 16.8M | 58.8 | **44.9** | 79.8 | 72.5 | 76.4 | **64.1** | 97.3 |
| Concat-Att (4L) | 12.6M | 59.0 | 45.0 | 81.1 | 71.8 | 76.9 | 64.0 | 98.0 |
| Cross-Att (2L) | 8.4M | 58.0 | 43.9 | 79.2 | 70.4 | **77.5** | 62.9 | 96.6 |
| Concat-Att (2L) | 6.3M | 57.7 | 43.8 | 79.8 | 70.0 | 76.9 | 60.3 | 97.1 |
| IPRM | 4.4M | **59.3** | **44.9** | **81.6** | **73.9** | 77.1 | 63.2 | 98.5 |

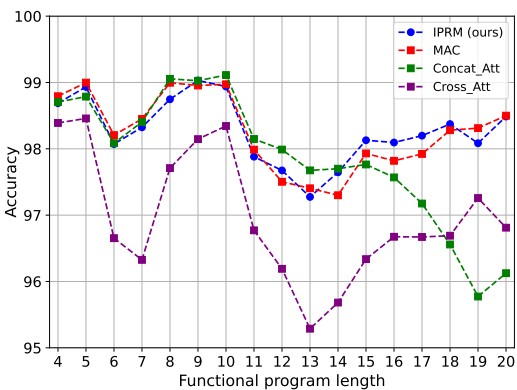

Figure 4: CLEVR accuracy vs functional program length (a proxy for question complexity).

Table 2: Performance on CLEVR-Humans and NLVR. IPRM exhibits strong zero-shot generalization on CLEVR-H and achieves state of the art on NLVRv1 when finetuned from CLEVR-Humans.

| Mechanism | CLEVR-Hmn | | NLVRv1 | |
|---|---|---|---|---|
| - | ZS | FT | Scrth | FT |
| FILM | 56.6 | 75.9 | 61.2 | - |
| MAC[a] | 58.7 | 78.9 | 59.4 | 69.9 |
| Cross-Att | 58.8 | 76.1 | 58.3 | 59.1 |
| Concat-Att | 59.3 | 78.7 | 57.4 | 65.0 |
| IPRM | **60.4** | **80.3** | **63.0** | **73.3** |

[a]refer to appendix regarding issues in reproducing MAC's reported performance on CLEVR-Humans

blocks. The number of heads for transformer blocks is set to 8. For MAC, on GQA we use a 4-step network (the optimal setting as per (Hudson & Manning, 2019b)), while for CLEVR, we use a 12-step network (as in Hudson & Manning (2018)). For IPRM, on GQA, we set the parallel operations to 6, iterations to 4 and window length to 2, and for CLEVR, were set them to 6 and 9 respectively. All mechanisms have dimension of 512 and recieve the same lang. and vis. inputs for fair comparison.

As shown in table 1, on GQA, IPRM achieves the highest performance of 59.3% while having only 4.4M parameters. Both 4-layer cross attention and 4-layer concat attention reach around 59.0%; however, they have substantially higher number of parameters (about 3.8x more for cross-att and 2.86x for concat-att). The 2-layer cross- and concat-attention have more than 1.3% lower peformance than IPRM while still requiring 1.4x to 1.9x more parameters. This suggests that pure feedforward parallel computation as in cross- and concat-att may require substantially higher parameters than a mechanism such as IPRM that integrates both parallel and iterative computation. On the other hand, MAC has a comparatively lower performance of 57.6% even though it has a higher number of computation steps than IPRM, and marginally higher number of parameters. This suggests that a pure iterative reasoning mechanism may not be able to effectively capture or maintain necessary vision-language interactions to answer the question.

The breakdown of performance by question type provides further insights into different capabilities of each reasoning mechanism. All mechanisms achieve similar performance on the "Query" questions. For "Verify" (where a condition has to be checked as yes/no) and "Logic" (where conditions have to be composed logically), IPRM does better than other mechanisms, notably having a 5.2% better performance than MAC for "Logic" questions. This suggests that perhaps the parallel operations of IPRM enable it to better retain and compose features in order to process "Logic" or "Verify" based questions. For "Choose" and "Compare", the both cross- and concat-attention mechanisms perform better than IPRM and MAC suggesting that transformer-based parallel computation may be more beneficial for such selection or comparison based questions. However, note that IPRM is about 0.4% and 0.9% lesser in these categories than the best model; in comparison, MAC is 3.5%

and 4.7% lesser. This again suggests that the parallel operation in IPRM is able to help it capture these reasoning capabilities to a decent degree.

On CLEVR, MAC obtains the best result of 98.9%, while IPRM achieves 98.5% and the best transformer-attention reaches 98.0%. These suggest that MAC's 12-step iterative computation allows it to better perform compositional reasoning for the entailed scenarios. To analyze how mechanisms perform over varying question complexities, we plotted their performance against the annotated functional program length in CLEVR (used as a proxy for question complexity). As shown in fig. 4, both MAC and IPRM perform consistently decently across both short- and long-functional program lengths. However, the Concat-Att mechanism's performance sharply drop after program length of 15, suggesting that the lack of iterative reasoning limits its abilities to process questions with multiple intermediate steps.

## 3.2 Reasoning Generalizability, Task Transfer and Compositional Generalization

In this experiment, we studied how well models generalize to new reasoning scenarios and their capabilities to transfer to different tasks. For this, we considered CLEVR-Humans wherein questions are free-form human generated. We also considered NLVR for language grounded visual reasoning and FILM (Perez et al., 2018) – a prominent visual reasoning mechanism. As shown in table. 2, we find that a IPRM model trained on CLEVR when tested zero-shot on CLEVR-Humans achieves 60.4%, which is better than the other mechanisms (particularly 1.7% more than MAC) as well as prior state of the arts (59.9% by MDETR (Kamath et al., 2021) – which uses auxiliary phrase-localization pretraining with a finetuned vision encoder and larger Roberta language backbone). In comparison, MAC had lower performance than both cross- and concat-att mechanisms (of 4Layers), even though it reached state-of-the-art on CLEVR. This suggests that MAC's iterative processing mechanism may over-tune to CLEVR's training distribution, and thereby achieve lower generalization on CLEVR-Humans. We also tested when models were finetuned on CLEVR-Humans, and found IPRM achieves 80.3% performance, again better than baselines.

For NLVR, we tested models that were trained from scratch and also finetuned (initialized with their best CLEVR-Humans checkpoints). Note, that NLVR is a distinct task from CLEVR and requires models to output True/False given a statement about three images. As shown in the same table, we found IPRM exhibits much stronger performance than baselines, both when trained from scratch and when finetuned. It particularly does 3.6% and 3.4% better than MAC, and obtains a performance of 73.3% when finetuned outperforming the previous state-of-the-art CNN-BiATT which achieves 66.1%. This shows that IPRM exhibits strong task transfer abilities. We also find that both Cross-and Concat-Att mechanisms perform poorly in comparison to IPRM and MAC suggesting the importance of iterative computation in language-grounded visual reasoning.

Finally, we also tested models on the CLEVR-CoGen dataset wherein in condition A (condA), shapes have a particular set of colors, while in condition B (condB), the shapes each have a distinct set of colors (e.g. red cube, blue sphere in condA; red sphere, blue cube in condB). This tests the ability of models to perform disentangled feature learning and learn primitive concepts separately. As shown in table 3, we find that Cross-Att achieves the best performance of 78.8% when trained in condA and tested on condB with IPRM close behind at 78.4%. Meanwhile, while MAC has a high performance of 98.5% on condA, it has poor performance on condB (2.5% lesser than IPRM). This suggests that MAC may have to learn feature conjunctions due to purely iterative reasoning whereas parallel mechanisms may process features separately. When all models are then finetuned on condB, we find IPRM obtains performance of 95.1% on condA and 97.0% on condB while the cross-attention mechanism fails to retain information from its original training.

## 3.3 Ablative analysis and Computation Visualization

In table 4, we perform ablations to study components of our mechanism. First, we study the impact of varying number of parallel operations M and computation steps T. We find that both (M=6, T=9) and (M=9 and T=6) perform the best, and that increasing M beyond 6 does not seem to benefit. We also see the impact of window size and find that a window size of 2 works best. Finally, we study the impact of removing the memory composition unit, and find drops of 6.2% on CLEVR and 3.6% on NLVR. However, on GQA the drop is relatively less (2.1%) possibly as it requires less composition.

Table 4: Ablations

| M | T | | |
|---|---|---|---|
| | 3 | 6 | 9 |
| 3 | 98.1 | 98.1 | 98.4 |
| 6 | 97.2 | 98.5 | 98.5 |
| 9 | 97.8 | **98.6** | 98.5 |
| Window Size | | | |
| 0 | 1 | 2 | 3 |
| 98.2 | 98.2 | **98.5** | 98.3 |
| Memory Composition Unit | | | |
| | CLEVR | GQA | NLVR |
| without | 92.3 | 57.1 | 59.4 |

Table 3: Performance on CLEVR-CoGen

| Mechanism | Trn Cond. A | | FT Cond. B | |
|---|---|---|---|---|
| - | Val A | Val B | Val A | Val B |
| FILM | 98.3 | 75.6 | 80.8 | 96.9 |
| MAC | **98.5** | 75.9 | 94.2 | **97.2** |
| Cross-Att | 97.4 | **78.8** | 92.8 | 95.6 |
| Concat-Att | 98.0 | 77.7 | 88.6 | 96.7 |
| IPRM | 98.3 | 78.4 | **95.1** | 97.0 |

We provide step-by-step reasoning of our model in A.1. Across the time steps, each parallel operation in the model seems to capture different aspects of language and relevant visual locations. In the initial few computation steps, the model appears to be computing the operation for ''*maximally occurring color''* and in the final step, finds the correct shape and makes the prediction. The visualization framework will be provided in source code.

## 4 RELATED WORK

**Visual reasoning and vision-language models and interaction mechanisms.** Multiple prior works have looked into the development of more effective visual reasoning mechanisms (Johnson et al., 2017b; Mascharka et al., 2018; Andreas et al., 2016). Prominent works include FILM (Perez et al., 2018), Neural State Machine(Hudson & Manning, 2019a), MAC (Hudson & Manning, 2018), Neuro-Symbolic-Concept Learner(Mao et al., 2019) and GAMR (Vaishnav & Serre, 2022). In contrast to these works that show applicability of models for particular tasks, our work explores a more general direction of integrating parallel and iterative computation in a reasoning framework that we show to be suitable for multiple visual reasoning tasks. More recently, vision-language models use transformer-based mechanisms showing impressive reasoning capabilities at scale. Notable examples include MDETR (Kamath et al., 2021), BLIP (Li et al., 2022; 2023), Flamingo (Alayrac et al., 2022) and OFA (Wang et al., 2022). We believe our work is complimentary to these developments, as it contributes an alternative and possibly more effective mechanism to the traditional cross and concat-attention methods, and could be effectively integrated with these models.

**Memory and recurrence-augmented transformers.** Multiple works have identified the limitations of purely feedforward computation as realized in transformers and worked in encoding recurrence (Hutchins et al., 2022; Huang et al., 2022; Dehghani et al., 2018) and memory-augmented computation (Wu et al., 2022; Bulatov et al., 2022). For example, the Recurrent Memory Transformer (Bulatov et al., 2022) introduces memory and recurrent computation to improve language modelling capabilities at smaller scales, while MemViT(Wu et al., 2022) introduces a cache-based memory to effectively retain prior context for long-video tasks at smaller scales. While these methods show the benefit of recurrent and memory-augmented computation across different tasks, our work focuses on the integration of memory and iterative computation in the context of visual reasoning, to develop a more powerful iterative and parallel reasoning mechanism.

## 5 CONCLUSION

We propose a novel mechanism for visual reasoning tasks that combines the benefits of both iterative and parallel computation. Our lightweight mechanism exhibits stronger reasoning capabilities and generalization than existing recurrent as well as transformer-based reasoning and vision-language interaction mechanisms while requiring less number of parameters and computation steps. Combining parallel and iterative computations obtains state-of-the-art zero-shot performance on question answering in unseen scenarios (Clever-Humans) and visually grounded natural language questions (NLVRv1). The reasoning steps of our proposed mechanism can be visualized step-by-step which aids interpretability and understanding.

## REPRODUCIBILITY STATEMENT

Source code and pre-trained models will be released publicly via Github upon acceptance. We have used three random seeds and whenever possible we have repeated the experiments at least three times to enhance the reproducibility.

## ETHICS STATEMENT

The real-world datasets used for visual reasoning may be biased based on their collection and annotation. These biases may be exhibited in the learned reasoning models as well.

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

# A    APPENDIX

Table 5: Collated results across benchmarks and comparisons with state of the arts. *NSM(Hudson & Manning, 2019a) uses scene graph supervision. ▼ NS-VQA(Yi et al., 2018) and TbDNet (Mascharka et al., 2018) use predefined dataset-specific (CLEVR) programs or modules. ■ MDETR(Kamath et al., 2021) performs auxiliary phrase-localization pretraining with a finetuned vision encoder and Roberta (Liu et al., 2019) language backbone.

| Module | Param. | GQA | CLEVR | CLEVR-Hmn | | NLVR | | CoGen-Tr-A | | CoGen-FT-B | |
| --- | --- | --- | --- | --- | --- | --- | --- | --- | --- | --- | --- |
| - | - | testdv | - | ZS | FT | Scrtch | Hmn-FT | Set.A | Set.B | Set.A | Set.B |
| Cross-Att (4L) | 16.8M | 58.8 | 97.3 | 58.8 | 76.1 | 58.3 | 59.1 | 97.4 | **78.8** | 92.8 | 95.6 |
| Concat-Att (4L) | 12.6M | 59.0 | 98.0 | 59.3 | 78.7 | 57.4 | 65.0 | 98.0 | 77.7 | 88.6 | 96.7 |
| Cross-Att (2L) | 8.4M | 58.0 | 96.6 | 58.2 | 75.1 | - | - | - | - | - | - |
| Concat-Att (2L) | 6.3M | 57.7 | 97.1 | 57.8 | 76.6 | - | - | - | - | - | - |
| FILM (Perez et al., 2018) | 3.3M | - | 97.7 | 56.6 | 75.9 | 61.2 | - | 98.3 | 75.6 | 80.8 | 96.9 |
| MAC (Hudson & Manning, 2018) | 5.8M | 57.6 | 98.9 | 58.7 | 78.9 | 59.4 | 69.9 | 98.5 | 75.9 | 94.2 | **97.2** |
| Benchmark-SOA | - | **61.6*** | **99.8**▼ | 59.9■ | **81.7**■ | **66.1** | - | **99.8**■ | 76.7■ | 96.9▼ | 96.3▼ |
| IPRM | 4.4M | 59.3 | 98.5 | **60.4** | 80.3 | 63.9 | **73.3** | 98.3 | 78.4 | **95.1** | 97.0 |

## A.1    FULL COMPUTATION VISUALIZATIONS

# B    IMPLEMENTATION DETAILS

**CLEVR**: For fair comparison with existing methods, we used pre-extracted ResNet101 features of 1024x14x14 with additional 2d positional embeddings. We utilized an LSTM backbone to process language inputs and the embedding layer and tokenizer were initialized with a BERT embedding layer. All models were trained for 30 epochs for CLEVR and CLEVR-Cogen and checked for convergence. For CLEVR-Humans, models were finetuned from their CLEVR checkpoint for 40 epoch. We utilized a learning rate of 1e-4 and Adam optimizer Kingma & Ba (2014). For transformer models, warmup was done on the first epoch starting from 0.0. Additionally, we found the best configuration of transformer models were 8 heads which we used for our experiments.

**GQA**: We used pre-extracted object proposal features from VinVL Zhang et al. (2021), and for each input integrated bounding box features and object label predictions along with the original bounding box features to form a joint visual representation. All models were trained for 25 epochs, and checked for convergence.

**NLVRv1**: Similar to CLEVR, we pre-extracted frozen ResNet101 features for each image (an NLVR image contains 3 images together in 1 file – these were split into 3 different files). A unique position embedding for each image was added when inputting 3 images to the model. Models were trained for 40 epochs for scratch and finetuned condition.

# C    REPRODUCING MAC PERFORMANCE ON CLEVR-HUMANS

Our reproduced MAC performance on CLEVR-Humans finetuning is 78.9%, which is lesser than MAC's reported performance of 81.5%. We tried multiple runs to see if better results are obtained but were unable to do so. Further, we note that similar issues have been reported in MAC's official codebase (https://github.com/stanfordnlp/mac-network/issues/30) where another reproduced performance is 76.6%.

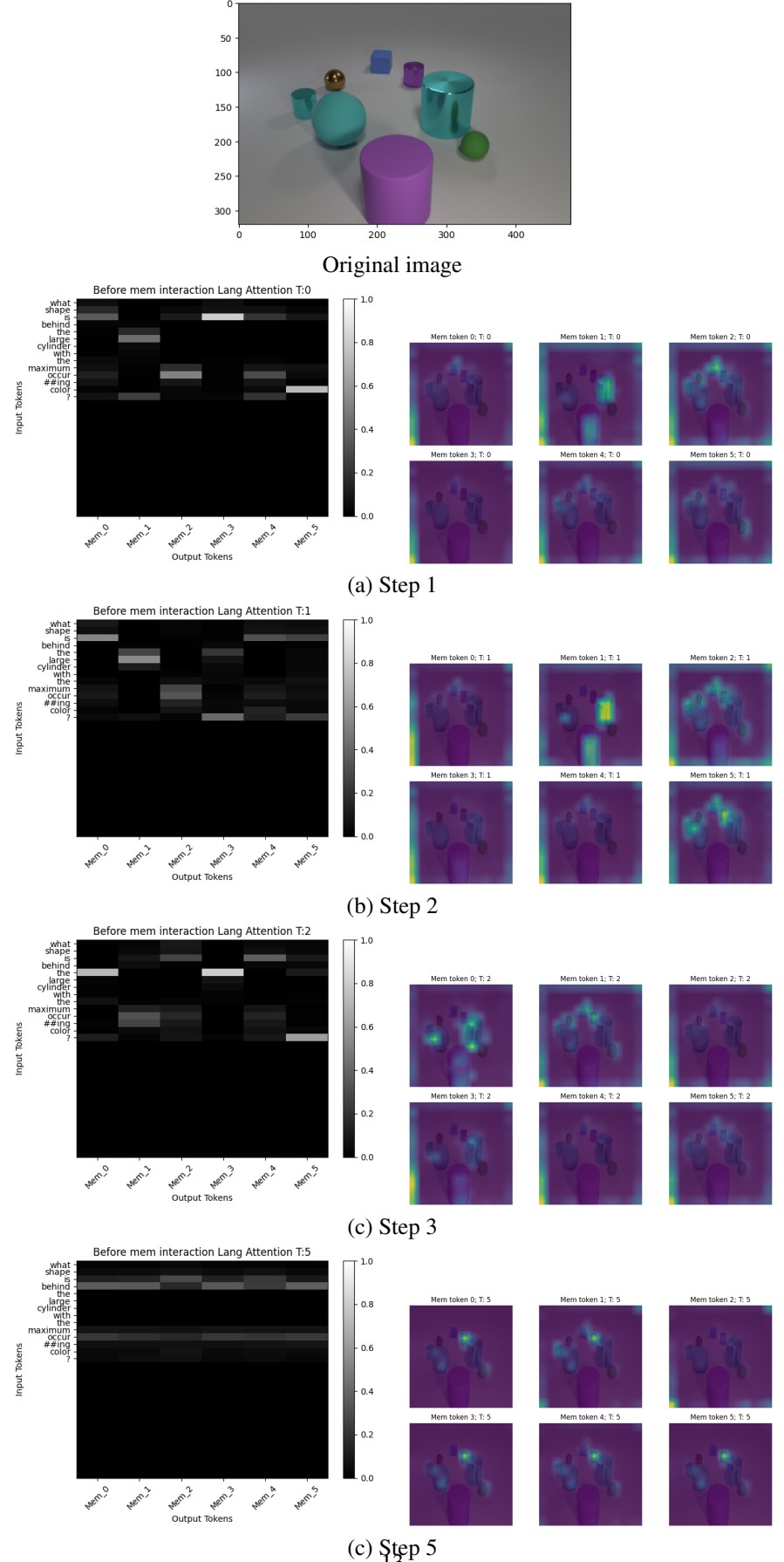

Figure 5: Step-by-step visualization of reasoning in IPRM. The question is: "What shape is behind the large cylinder with the maximum occuring color?" and prediction is Cylinder

