# OpenReview forum: "Learning to reason iteratively and parallelly for complex visual reasoning"
_ICLR.cc/2024/Conference — ICLR 2024 Conference Withdrawn Submission_

### Official Review · Reviewer_kvgs · 2023-10-30

**Soundness:** 3 good
**Presentation:** 2 fair
**Contribution:** 3 good
**Rating:** 5
**Confidence:** 3

**Summary:**

A new neural network-based visual reasoning pipeline is proposed that processes the language and visual tokens both parallelly and iteratively. The architecture is modular and consists of three parts that happen iteratively at each time step -- (a) Operation Formation where attention mechanisms are used between the input language tokens and existing operation states to form the operation vectors, (b) Operation Execution where the cross-attention between the operation vectors and the visual tokens yield the operation result vectors and (c) Operation Composition where the the operation vectors in memory are composed with other operation vectors at the current time step as well as prior operation states to form the states for the next time step using more cross attention mechanisms. The result of the reasoning process is then obtained using another cross-attention layer between the language vectors and operation result states to yield the final result. Experiments are performed on many standard visual reasoning datasets that test compositional visual question answering, generalization etc. Compared to simple cross-attention, concat attention and MAC (also a recurrent scheme), the results from the proposed method are favorable overall.

**Strengths:**

1. The motivation for the architecture is interesting and reasonable.
2. The modular nature of the reasoning pipeline makes the overall steps seem intuitive.
3. The experiments show improved performance on several benchmarks over other methods that do not use explicit parallel / iterative reasoning.

**Weaknesses:**

One of the main things that needs to be addressed is the part of the architecture that does the parallel reasoning. From the architecture description, this seems to arise of the first dimension in the some of the internal state matrices of dimension $N_m$. Is it due to the masks in the Operation Composition stage that make these operations "parallel" so that these internal states attend to different parts of the input parallelly? I am trying to understand the difference between having a larger internal state versus explicit parallel reasoning. Where is the structure enforced in the architecture that requires parallel processing when possible?

Related to the above, there is one line that says identity attention is used to "enable a higher degree of operation composition". I don't follow this and needs clarity. Overall, I am not sure how this operation composition step is different and not just implicitly subsumed into earlier architectures, where is the parallel processing explicitly arising from?

I think the iterative nature, on the other hand, is immediately due to the recurrent nature of the algorithm.

There seem to be many design decisions that are not immediately clear as being intuitive, and need further explanation. This is true of the Operation Composition stage. It seems to me that this stage is supposed keep a record of the operations performed up to a time step. It is said that operation states are composed with states from multiple time steps in the past. I don't follow this, isn't the M matrix already encoding memory of states in some way, why look back multiple time steps? This stage of the architecture is difficult to follow.

The visualizations in the supplement for one example seem to be pointing to some interesting aspects of the algorithm. But I don't see where the parallel reasoning is being utilized or even the iterative aspects. I think a clear explanation of the figure is needed, I don't understand how to interpret the memory states, the parallel processing, or how the prediction "cylinder" came into being. Perhaps visualizing other matrices and attention masks is also important here.

Many more visualizations should be added to illustrate these important parts of the algorithm. Otherwise, it is hard to say whether it is indeed the parallel and iterative structure that is helping or a different issue.

Some of these questions also arise because the numbers themselves are also hard to interpret. I don't see why test samples for "logic" type in Table 1 need parallel computation. The reasoning the authors provide are circular and seem to be based on the numbers themselves.

**Questions:**

A minor question, is $N_m$ the same as M in Table 4?

---

### Official Review · Reviewer_iiFw · 2023-10-31

**Soundness:** 3 good
**Presentation:** 3 good
**Contribution:** 3 good
**Rating:** 6
**Confidence:** 2

**Summary:**

The paper introduces Iterative and Parallel Reasoning Mechanism (IPRM) that is able to combine iterative and parallel operations and benefit from the advantages of both. Authors perform various experiments and show the performance of their approach compared to baseline approaches on state of the art benchmarks along with its generalization ability.

**Strengths:**

1. The paper is written clearly and is easy to follow.
2. The motivation behind the work is clear.
3. Reasonable number of baselines are considered along with datasets.

**Weaknesses:**

1. For some results their method lags behind some of the SOTA approaches (both for the main results as well as the generalizability results).
2. For the ablation study it would have been better if the results were discussed in more detail and more insights were provided to the reader.
3. The paper claims that the introduced approach is lightweight requiring less number of parameters and computation steps. It would have been good if some numbers were provided or some ablations or studies to prove this point further.
4. It would have been better of some qualitative examples were provided for the results sections on types of cases where the approach would fail or would work for the reader to get more insights.

**Questions:**

Some of my questions are listed in the weaknesses (particularly 3 & 4).

---

### Official Review · Reviewer_bPrS · 2023-11-01

**Soundness:** 3 good
**Presentation:** 3 good
**Contribution:** 2 fair
**Rating:** 5
**Confidence:** 4

**Summary:**

The paper addresses visual reasoning problems. It introduces a fully neural iterative and parallel reasoning mechanism (IPRM) that combines iterative computation with the ability to perform distinct operations simultaneously. Evaluation is performed on visual reasoning datasets such as CLVER and GQA. IPRM achieves state-of-the-art zero-shot performance on the CLEVR-Humans dataset.

**Strengths:**

• The proposed method is novel and interesting.
• The paper is well written and easy to understand.
• The paper includes adequate ablations in Table 4.

**Weaknesses:**

• A big drawback of the proposed IPRM model architecture seems to be that it cannot be trained in parallel unlike transformer-based models. A detailed comparison of training times with transformer based models with similar number of parameters is necessary.

• Scaling model size: on GQA IPRM uses around 4.4M parameter. How does the performance change if the number of parameters is increased/decreased? Transformers generally scale well with respect to performance with increasing number of parameters (if properly regularized).

• Performance of IPRM is consistently below the state of the art: In GQA “Coarse-to-Fine Reasoning for Visual Question Answering, CVPR Workshops 2021” achieves 72.1% compared to 59.3% reported for IPRM. In CLEVR, MDETR “Modulated Detection for End-to-End Multi-Modal Understanding, ICCV 2021” achieves 99.7% accuracy compared to 98.5% accuracy reported for IPRM. In CLEVR-Humans MDTER achieves  81.7% accuracy compared to 80.3 reported for IPRM.

• Inference speed: compared to transformer architectures IPRM is more involved with matrix operations. A detailed comparison of inference speeds with respect to plain transformers with a similar number of parameters is necessary.

**Questions:**

• More detailed comparison of training efficiency and inference speeds in comparison to pure transformer based approaches is necessary.
• The paper should better motivate the reason for weak performance on CLEVR and GQA.

---

### Official Review · Reviewer_Mmxk · 2023-11-04

**Soundness:** 4 excellent
**Presentation:** 3 good
**Contribution:** 3 good
**Rating:** 5
**Confidence:** 4

**Summary:**

This paper introduces a novel mechanism to combine iterative and parallel computation to solve visual reasoning tasks. The model takes visual and language features (describing the tasks) and outputs a reasoning result. Then a memory state keeps track of the operations taken over and its updated with the results tokens obtained from previous operations. The model is tested on several visual reasoning tasks such as CLEVR-humans where it obtains state of the art results.

**Strengths:**

The paper presents a novel perspective on integrating iterative and parallel computations for reasoning tasks. It is well-crafted and compellingly motivated. The model's performance on visual reasoning tasks suggests that it not only competes well across various benchmarks but also exceeds the current state of the art in certain scenarios. In my view, the paper successfully amalgamates multiple intriguing mechanisms to attain excellent results. I value the authors' inclusion of clear diagrams to elucidate the computational process and their detailed discussion of each mechanism involved.

**Weaknesses:**

My primary concern with the paper is the presentation of the results. I suggest some reorganization for a more reader-friendly structure, allowing clearer navigation through each experiment and its corresponding results. For instance, Tables 3 and 4 lack captions, which are essential for context. Moreover, Tables 1 and 2, as well as Figure 4, disrupt the flow of reading because they are not referenced until much later in the text. The step-by-step reasoning depicted in Figure 5 appears interesting; however, it lacks a clear description of its implications. Overall, I believe that reorganizing the results and discussion sections could significantly enhance the paper's clarity and readability.

**Questions:**

I believe the paper could have provided more insight in the results section regarding how the two modules—iterative and parallel—interact during the inference phase. The visualization in Figure 5, for instance, did not make it clear whether the iterations were enhancing the refinement of the solution, nor the role of the parallel stream. Also, the reason for the high activity at the border of the memory token was not explained.

I felt the need for additional illustrations on why the iterative steps are necessary for answering complex questions. While Figure 4 shows some evidence that the model performs well with questions of varying lengths, I would have liked to see a graph detailing the model's performance across a range of time-step budgets.

Furthermore, the claim in Section 2.5 that "the IPRM is end-to-end trainable, and can directly learn to flexibly optimize the number of parallel and iterative computations performed, as well as the extent to which each contributes to the final reasoning output," should be supported by experimental results that demonstrate this capability.